# Influence of Curing Temperature on the Strength of a Metakaolin-Based Geopolymer

**DOI:** 10.3390/ma16237460

**Published:** 2023-11-30

**Authors:** Adelino Lopes, Sérgio Lopes, Isabel Pinto

**Affiliations:** 1INESC Coimbra, Department of Civil Engineering, University of Coimbra, 3030-290 Coimbra, Portugal; avlopes@dec.uc.pt; 2CEMMPRE, ARISE, Department of Civil Engineering, University of Coimbra, 3030-788 Coimbra, Portugal; 3Department of Civil Engineering, University of Coimbra, 3030-788 Coimbra, Portugal; isabelmp@dec.uc.pt

**Keywords:** geopolymer, metakaolin, alkaline activation, curing temperature, mechanical strength, flexural strength, compressive strength, stress-strain behavior

## Abstract

The present work focuses on the further development of a new family of geopolymers obtained by the alkaline activation of a binder. The aim is to find a viable alternative to concrete that can be used in civil construction. Regarding the influence of the curing temperature on this type of mixture, the recommendations in the existing literature are different for fly ash, ground granulated blast-furnace slag, and metakaolin-based geopolymers. While for fly ash and slag, increasing the curing temperature above 60 °C is reported to be advantageous, for metakaolin geopolymers, the opposite is reported. In this context, the objective of this work is to evaluate the mechanical strength of several metakaolin-based geopolymer specimens subjected to different curing temperatures (10, 15, 20, 30, 40 and 50 °C). Furthermore, several stress-strain diagrams are also shown. Based on the results, we recommend using curing temperatures below 30 °C in order to avoid reducing the strength of metakaolin-based geopolymers. Curing at 50 °C, relative to room temperature, results in a reduction of more than 35% in flexural strength and a reduction of more than 60% in compressive strength. Regarding the behavior of the geopolymers, it was found that the strain, at the ultimate stress, is about 2 to 2.5 times the strain of an equivalent cement mortar.

## 1. Introduction

To reduce carbon emissions, there is a worldwide effort to provide a standardized and systematic approach for measuring and reporting greenhouse-gas emissions and to support efforts to reduce emissions, mitigate climate change and improve sustainability. Some organizations, such as the Carbon Trust Standard (https://www.carbontrust.com/en-eu, accessed on 2 September 2023) and the Green House Protocol (https://ghgprotocol.org/, accessed on 2 September 2023), also have specific standards for measuring and reporting the CO_2_ footprint of products and services.

In this context, one cannot fail to mention the construction sector, specifically the part that systematically and preferentially uses structural concrete. It is known that construction is highly dependent on the use of natural ores and those derived from fossil fuels. The most significant of these is obviously Ordinary Portland Cement (OPC), the material most used in civil construction in the developed world. One of the major limitations of using OPC is its manufacturing method, which is highly demanding in terms of energy and therefore highly polluting in terms of CO_2_. Finding an alternative to concrete involves developing a material that (i) has equivalent strength and mechanical properties; (ii) can be built cost-effectively; (iii) exists in sufficient quantity in the world. Unless these three initial requirements are met, it will be difficult to make the civil construction market accept the new material widely.

Geopolymers, also known as alkali-activated binders, are presented as an alternative to ordinary Portland cement (OPC), with inherent benefits such as sustainability, increased durability, and the capacity to incorporate various residues [1,2]. Positioned as a third-generation cement, succeeding lime and OPC, geopolymers are alkali aluminosilicates referred to by various names, including inorganic polymers, alkali-activated cements, geocements, alkali-bonded ceramics, and hydroceramics—all operating on the same chemical principles [3]. Geopolymers are commonly produced using fly ash (FA) [4,5], ground granulated blast-furnace slag (GGBFS) [6,7], metakaolin (MK) [8,9], or a combination of these binders [10,11,12], with the option of including one in the mix [13,14,15]. Additionally, contemporary formulations explore alkaline activation of other minerals and industrial byproducts rich in aluminosilicates.

The geopolymerization process using FA or GGBFS binders is significantly influenced by curing temperature and the duration of exposure to temperature [4,16,17]. Curing temperature is crucial for developing the requisite strength, but it imposes limitations on the material’s environmental friendliness (in terms of CO_2_ emissions) and its suitability for structural applications. An additional drawback associated with FA- and GGBFS-based geopolymers is shrinkage. In contrast, MK-based geopolymers have reasonable strength for civil construction when cured at ambient temperature and do not experience the same level of shrinkage [18,19].

The objective of this study is to assess the practicality of using geopolymers made by alkaline activation of MK as a sustainable substitute for concrete in structural applications. As an exploratory investigation of the mechanical characteristics of geopolymers, this research aims to evaluate the mechanical strength of several MK-based geopolymer specimens subjected to various curing temperatures (10, 15, 20, 30, 40, and 50 °C). Importantly, these materials do not require additional energy for curing after mixing, thus aligning with the goal of producing eco-friendly materials. Additionally, the stress-strain diagrams of several specimens cured at ambient temperature are presented.

In contrast to the conventional approach, there is no intention to enhance the strength characteristics of the geopolymer in this study. The primary objective is to address a question that emerged during the pouring of MK geopolymer beams. Specifically, it was observed that the surface temperature of the beams reached around 47 °C [20]. This temperature exceeded that which the authors had previously witnessed during the curing of small samples or concrete masses. The key question that arose was whether this temperature could affect the strength characteristics of the geopolymer. Consequently, the study involves testing various temperatures over approximately 50 h to provide insights into this question.

## 2. Brief Literature Review

### 2.1. Concrete

The simplest form of concrete (plain concrete) is a mixture of OPC, water, and aggregates (such as sand and/or gravel). However, currently, concrete is produced in a much more complex way, with this term generally referring to several types of specialized concrete that are designed to meet specific construction needs or challenges. Examples include high-strength concrete, self-compacting concrete, fiber-reinforced concrete, lightweight concrete, etc. 

The term “OPC”, on the other hand, can refer to a range of binding materials that are used to create concrete. The most common type of cement used in concrete production is Portland cement. However, other types of cement exist. For example, the most common is Portland-composite cement [21], which is made by blending other binders (cementitious additives) with Portland cement. Examples of these binders include silica fume, FA, GGBFS, MK, etc. Additionally, chemical admixtures like accelerators, retarders, or plasticizers may be used to modify the properties of the concrete according to specific project requirements. Overall, special concretes can be tailored to meet specific project requirements and can offer advantages over traditional concrete in terms of performance and durability. However, they may also be more expensive and require specialized knowledge and expertise for production and placement.

As far as concrete is concerned, the required knowledge and standards exist, and the construction sector can use them globally. A simple compression test allows the material to be quantitatively characterized from a mechanical point of view (using a stress-strain diagram) [22]. Thus, knowledge of the 28-day compressive strength of a concrete is enough to allow engineers to make calculations that will allow the construction of a structure that is safe. Of course, it is also necessary to ensure that this concrete will keep its mechanical strength during the whole life of the structure.

In terms of environmental impact, the total volume of OPC production worldwide has increased from ~1.4 Gmetric tons in 1995 to ~4.1 Gmetric tons in 2013. Production stabilized after that point [23]. Regarding the production of CO_2_, there are some discrepancies in estimates. For example, Torgal et al. [2] reported that one ton of CO_2_ is emitted for each ton of OPC. Tasiopoulou et al. [24] have reported 923 kg CO_2_-eq/ton. The CEMBUREAU (www.cembureau.eu, accessed on 2 September 2023), in its 2021 activity report, reported 783 kg per ton of OPC in 1990 and a target of 472 kg per ton of OPC by 2030. Strategies for mitigation include, for instance, the use of non-recyclable and biomass waste to replace fossil fuels. Some consensus on this matter is needed in order to evaluate the alternatives.

Two predictions for the future, made by Aïtcin [25] in 2000, are as follows: (i) “The concrete of tomorrow will be GREEN” and (ii) “Cement and concrete will remain, at least during the first half of the 21st century”.

### 2.2. Alternative to Concrete

From an environmental perspective, alternative materials that are more sustainable than OPC are being explored. Geopolymers are one such material that is gaining attention as a possible alternative to OPC. However, it is important to also consider the socio-economic factors involved, as well as technical concerns such as shrinkage, creep, fatigue, etc. Despite these technical challenges, the environmental issue remains a critical consideration.

According to Komkova and Habert [26], the global warming potential (accounting for CO_2_ eq. emissions) of alkali-activated material mixes (AAMs) is lower (39–57%) than that of OPC concrete mixes. This conclusion also holds when variability in the production of constituents for both AAMs and PC concrete and uncertainty associated with transportation distances are taken into account. The main sources of uncertainty are variability in the production of precursors and activators.

According to Palomo et al. [8], the mechanical strength of FA-based materials is significantly influenced by factors such as temperature and the type of activator used. For example, prisms cured at 85 °C exhibit much greater strength than those cured at 65 °C. Additionally, longer curing times generally lead to higher average strength. Several researchers [17,26,27] have confirmed these conclusions. However, such high-temperature curing requirements can limit the material’s environmental sustainability and impact its potential for application in structural elements for civil construction. It is worth noting that recent research [6,7,10,11,12] has explored curing specimens at ambient or room temperature as an alternative approach.

A second issue to be analyzed concerns the quantities available. The total volume of cement production worldwide amounted to an estimated 4.1 Gton in 2022 [23]. According to a report by the Global Coal Ash Research Network (GCARN), global production of coal FA (including both Class F and Class C FA) was estimated to be approximately 0.78 Gton in 2018. However, the pressure to close coal-fired power stations is enormous, particularly in Western countries. Therefore, these quantities will naturally be reduced for reasons of ecological sustainability. In addition, it is necessary to verify whether the FA really is a viable alternative. According to a report by the Global Slag Knowledge Base, global production of GGBFS (byproducts of steel production) was estimated to be around 0.36 Gton in 2018. With regard to the remaining cementitious additives (silica fume, volcanic ash pozzolans, etc.), the true quantities produced globally each year are not known, but it is known that they are produced in small quantities relative to FA and GGBFS. It is also known that USA cement companies are grinding of GGBFS [23]. A quick analysis of these values leads to the conclusion that these alternative binders replace only about 20% of the OPC. With regard to Portland-composite cements, this value agrees with those of the ACI 232.2R-18 [28] and EN 197-1 [21]. However, the amount of FA and GGBFS produced annually is insufficient to meet the global requirement for binders.

MK (another common supplementary cementitious material) could also be used to meet future demand for binders [29]. MK is a highly reactive pozzolanic material that is commonly used as a supplementary cementitious material in concrete. It is produced by calcining kaolin clay at a high temperature, which results in a highly reactive amorphous silica-and-alumina material [8,13]. Although its production requires energy, the energy required is much less than that used for OPC; according to Tasiopoulou [24], the requirements fall into the range of 463–695 kg CO_2_-eq/ton. This variability can be even greater; according to Rashad [30], “The optimum temperature for heating kaolin to obtain MK may be in the range from 600 °C to 850 °C for 1–12 h”. Despite being more expensive than other industrial byproducts, MK offers technical and environmental advantages [13] and can be found around the world and in large quantities [31]. It is also important to mention a substantial advantage of MK in relation to other industrial by-products such as GGBFS or FA. According to Duxson et al. [32], “metakaolin-based geopolymers can be manufactured consistently, with predictable properties both during preparation and in property development. … In contrast, fly ash is an industrial waste that is not derived from a well-defined starting material. … The particles in fly ash are generally spherical, but inhomogeneous, and comprise glassy as well as crystalline (often mullite and quartz) phases. The particle size distribution can be very broad, and different size fractions will differ in elemental and phase composition. This degree of inhomogeneity means that more care is required when working with fly ash to ensure that the optimal mix design is obtained for a given ash if a consistent product is to be obtained”. This issue is crucial for materials used in the construction industry.

For the construction sector, it is important that the alternative material has equivalent mechanical strength, is available on a large scale, has equivalent costs, and is known to current workers, namely engineers. Furthermore, the manufacturing must be regulated in a manner equivalent to that applied to concrete manufacturing; materials must be of the quality specified in regulations; and the mechanism of assessing that quality (via tests) must also be equivalent. In this regard, two details are particularly significant. (1) In the overwhelming majority of articles relating to geopolymers, XRD (X-ray diffraction analysis), and/or SEM (scanning electron microscopy), and/or XRF (qualitative X-Ray fluorescence), and/or IR (infrared spectroscopy), and/or EDX (energy-dispersive X-ray spectroscopy) and similar tests are used in order to assess the characteristics of geopolymers. In fact, when developing the material (improvement of strength characteristics, impermeability, etc.), these tools are important because they make it possible to qualitatively extrapolate these quantities. However, in the construction phase, the way to characterize (quantitatively) the strength of the material (for example) is to use regulated tests. (2) Another characteristic discussed in specialized articles is the quantification of mixes in terms of the molar ratios of SiO_2_ to Al_2_O_3_ and Na_2_O to Al_2_O_3_ [33,34,35] and other oxides. Quantification of binder composition in terms of oxides is also performed (almost articles on this topic). The need for such quantification is clear when dealing with reduced samples of a material that is unknown in the market. However, construction usually deals with materials that are generic and whose composition is known from the manufacturer. Identical reasoning must be applied to the remaining components of mixtures. Incidentally, there are cases in which the additives used can be purchased on the market, without the manufacturer specifying their chemical composition.

### 2.3. Shrinkage

Hardened concrete experiences strain over time due to compressive stress or load-free conditions, with strains encompassing elastic deformation (instantaneous strain), creep deformation (strain over an extended period), and shrinkage deformation (strain measured on load-free specimens) [36]. The total shrinkage deformation (TSD) comprises autogenous shrinkage (AS) and drying shrinkage (DS). Geopolymers from waste or industrial byproducts exhibit negative TSD [37,38], limiting their use in civil construction. In contrast, the concrete typically used in construction has a TSD range of 0.1–0.5‰ [39,40]. Lower values are acceptable, while higher values may negatively impact appearance and long-term deformation.

Unless the relative humidity (RH) falls within the range 90–100%, the total shrinkage deformation (TSD) is assumed to be similar to drying shrinkage (DS), as autogenous shrinkage (AS) is relatively low and is included in chemical shrinkage. RH plays a crucial role in TSD evaluation. However, there is notable inconsistency in the literature concerning DS values for geopolymers, with reported values ranging from 0.2‰ to 10‰ in various studies [41].

Table 1 presents some results taken from the bibliography. Based on the authors’ experience, the shrinkage level reported by Duran et al. [42] should be perceptible to the naked eye in any reinforced concrete beam, as the crack strain is lower than this value.

It should be mentioned that the authors have already built beams using MK-based geopolymers and that the huge retractions (>~1‰) reported above (Table 1) have never been observed.

It is important here to highlight two important conclusions of Perera’s work [34]: (i) “It is clear that the exposure to the RH oven atmosphere (30–70% RH), when the lid of the container is opened, causes a rapid drying of water which promotes cracking”; (ii) “To obtain geopolymers free of cracks, rapid drying during curing should be avoided by sealing the sample container”. For this reason, molds should be sealed with steel boards during solidification and hardening when curing above 60–70 °C is required. 

“Volume fraction solids” is another important parameter described by Kuenzel et al. [55] and Riahi et al. [56]. They reported that 40 vol% (Kuezel) or 50 wt% (Riahi) will be enough to control shrinkage in geopolymer mortars.

To conclude this point, it is crucial to note an unsuccessful attempt in which FA-based geopolymer was used without a curing temperature and plasticizer [37]. Following experiments with small specimens, the authors encountered significant material shrinkage when they applied the method to beams, rendering it unsuitable for civil construction. Unlike beams or slabs constructed with OPC or MK-based geopolymer, these beams underwent no cracking due to shrinkage, as observed by the authors.

### 2.4. Effect of Curing Temperature

This section of the report serves to underscore the uniqueness of this work by demonstrating that existing knowledge does not definitively address the question posed in the previous section.

Regarding the curing temperature, it is also important to define the temperature parameter. Test standards generally specify the curing temperature and the corresponding tolerance. In the structural field, the overwhelming majority of tests are carried out at room temperature, between 10 °C and 35 °C. Tests carried out under controlled conditions are carried out at 23 ± 5 °C.

In the particular case of curing concrete specimens, standard EN12390-2 [57] specifies a temperature of 20 ± 5 °C, or 25 ± 5 °C in hot climates, for a period of time between 16 h and 3 days. Among other things, this specification ignores the temperature inside the concrete mass during this process. That is, the specified temperature refers to the temperature of the air surrounding the specimens. Furthermore, the specified value is to be understood as an average value, i.e., the tolerance for the expected variation is not defined. In other words, the impacts of ambient temperature fluctuations around the mean value in the concrete mass are expected to be negligible.

Applying the same principle to this work, the authors expect that the temperature fluctuations of the surrounding air will have a negligible impact on the mass of the geopolymers during curing. This principle is valid for two additional reasons: first, none of the specimen surfaces were in direct contact with the air; second, the temperature cycles always had periods of less than 2 h. In this context, it is reasonable to assume that the curing temperature was almost constant. Obviously, the objective average temperature was never fully reached, so the results must be considered in relation to the calculated temporal average temperature.

It is important to start by mentioning that the parameter “curing temperature” is always associated with two other parameters that are no less important: the day of the test and the time interval over which the specimens were subjected to that temperature. Of course, the type and concentration of the alkaline activator can also be important. Many other parameters (use of aggregates, the effect of binder particle-size distribution, etc.) will not be addressed.

In this work, it is important to focus on the factor that is most important from a civil-construction perspective, i.e., on the mechanical strength of geopolymers.

Several authors have been studying the influence of curing temperature on alkaline-activated binders. Mainly in the case of FA-based geopolymers, it has been concluded that the strength properties improve when the curing temperature is much higher than room temperature [58,59,60,61,62]. These works led to three fundamental conclusions. Firstly, increasing the curing temperature improves the strength of geopolymers for the same curing period. Secondly, the strength values can be increased by increasing the curing time for all curing conditions. Third, the early strength of oven-cured geopolymers is clearly superior to the strength of air- or water-cured geopolymers. Most authors mention that the rate of the geopolymeric reaction increases with an increase in the temperature of the curing medium. On the other hand, as the duration of heat curing increases, FA-based geopolymer mixtures show higher polymerization compared to mixtures prepared with shorter curing times. The literature includes the following conclusions: the order of increase in strengths depends not only on curing temperature, but also on Na concentration; for low curing temperatures (~60 °C), 72 h of heat curing can be sufficient; for high curing temperatures (~85 °C), 24 h of heat curing can be sufficient; at very high curing temperatures (115 °C), no improvement in strength was observed; at shorter curing durations (~6 h), no significant improvement in strength was observed.

Regarding FA-based geopolymers, it is important to emphasize two points: the higher the curing temperature and the longer the curing time, the greater the mechanical strength of the specimens. However, there are consequences if one is trying to use this process in construction. In the first place, the applications of this process are limited to precast concrete structures only because curing at a high temperature (80 °C or so) is practically impossible in situ at large scales. Secondly, these two parameters (temperature and duration) contribute significantly to the energy costs of the process, i.e., the eco-friendly advantage of the product is lost.

The term “curing temperature” in the context of MK-based geopolymers needs to be reviewed. Two points can be highlighted: first, the number of investigators examining this parameter seems to be substantially lower than the number investigating other types of geopolymers; second, curing temperatures are substantially lower. 

We know that many mixing parameters, such as the thermal history of the source materials (ex., kaolinite, alkali concentration, initial solid content, etc.), have a substantial effect on the final properties of the geopolymer. In this study, we primarily consider the curing regime used for the geopolymer. The effect of curing temperature and duration on the development of the hard structure of MK-based geopolymers has been studied by many investigators [63,64,65,66,67,68]. Most of these studies found that curing conditions (temperature and duration) have a significant effect on the mechanical properties of MK-based geopolymers. Table 2 provides a comprehensive overview of the primary considerations and results within the context of this study, that is, this table pertains exclusively to MK-based geopolymers and those subjected to curing temperature. The highlighted aspects include the specific curing temperatures and time intervals under consideration. Subsequently, the table details the maximum compressive strengths attained through various processes and concludes with the main findings regarding the variations in strength.

The first curiosity in these studies is the curing temperature of 10 °C, a temperature that is not even considered for curing FA-based geopolymers. Furthermore, this curing condition is not an impediment to reaching the greatest strengths. For example, Lahalle [63] has concluded that in the longer term (90 days), samples cured at a lower temperature (10 °C) tended to have better performance than samples initially cured at a high temperature (30 °C).

In terms of general conclusions, it is possible to state that increasing the curing temperature will significantly accelerate the chemical reactions in the fresh mixtures, inducing rapid development of mechanical strength at early ages of curing relative to mortars cured at 20 °C. The main effect is that high strength develops sooner. However, long-term heat curing at too-high temperatures (such as 90 °C for seven days) likely reduces the mechanical performance of the MK-based geopolymer because the quick formation of the hard structure is unlikely to result in a good-quality product. Mo [64] observed that curing at high temperatures (>40 °C) has a detrimental effect on physical properties. The rapid setting speed of geopolymer slurries restricts their transformation into a compact, tough structure. Li [65] concluded that prolonged heat curing at excessively high temperatures (e.g., 90 °C for seven days) leads to pore coarsening and micro-defects in the gel phase due to chemical shrinkage, resulting in reduced mechanical performance. Arellano [67] noted that mortars cured at 75 °C exhibited greater proportions of unreacted particles compared to those cured at 20 °C. Additionally, the formation of finely distributed pores, approximately 10 μm in size, was observed, likely as a result of rapid water evaporation and fast binder densification due to increased curing temperature. Vitola [68] arrived at a similar conclusion, correlating increased porosity (and subsequent reduction in strength) with curing temperature.

Regarding the methodologies used, there are some aspects that are similar: for example, selected curing temperatures. However, there are several aspects that are significantly different, such as the curing time at that temperature (duration). Li [65] has adopted a seven-day curing time. Others [35,64] have selected shorter periods, with some curing the material for only hours. Curing at an elevated temperature (80 °C) for a shorter period (1 h) did not cause remarkable changes in strength, but longer curing (4 h) was responsible for a considerable acceleration of the reaction rate and an earlier increase in strengths. 

Nonetheless, Rovnaník [66] concluded that compressive strengths of materials cured for three, seven and twenty-eight days at 80 °C and at 60 °C are lower than the strength of materials cured at 40 °C. It is important to emphasize here that the alkaline reaction is exothermic. Therefore, at normal curing temperatures (20–40 °C), the geopolymer curing time is not less than three days and the curing temperature is not constant.

Mo et al. [64] concluded that “Appropriate elevation in curing temperature (below 60 °C) speed up the harden process and improve the physical properties of the geopolymer samples. However, curing at too high temperatures (80 and 100 °C) result in a negative effect on physical properties …”. This conclusion was drawn from tests performed for up to seven days.

Another curiosity evident in these works is the use of 28 days as a curing time. This time corresponds to the standard reference day after curing begins on which the concrete specimens are tested. The question that arises is whether this practice should be transferred to geopolymers. In the case of MK-based geopolymers, it is possible to conclude that it should not. For example, in the work by Lahalle [63], it is clear that after 14 days, the MK-based geopolymer has essentially reached its maximum strength. Indeed, the maximum strength was probably reached before that point. Considering the deformation of the material, which was equivalent to that attained on the 28th day of curing for an OPC-based concrete, Lopes et al. [69] have shown that metakaolin-based geopolymers reach maturity around day 15 or 16 of curing. Furthermore, and as stated above, increasing the curing temperature by a limited amount helps to reduce this time interval without reducing the final strength.

## 3. Materials and Methods

### 3.1. Material Preparation

Two types of metakaolin were used in this study: a white metakaolin commercially named “MetaMax^®^ HRM” (https://www.l-i.co.uk/products/metamax/, accessed on 2 September 2023) (referred to as MKW) and a brown metakaolin commercially known as “ARGECO” (referred to as MKB), which was provided by the French company “Argeco Développement” (https://www.argeco.fr/, accessed on 2 September 2023). An elemental analysis was conducted to study the chemical composition of these materials (Table 3). The table shows that the main components are silica (SiO_2_) and alumina (Al_2_O_3_). The density of MKW is 2500 kg/m^3^, with an average particle size of 1.2 μm and a specific surface area of 13 m^2^/g. MKB has a density of 2510 kg/m^3^, with an average particle size of 5.7 μm and a specific surface area of 20.1 m^2^/g.

Fine natural sand (collected in Coimbra, Portugal) with a particle size fraction of 0–4 mm was used as the aggregate in the production of the pastes. The density of the sand particles is 2.64 g/cm^3^. Based on particle-size analysis, the sand can be classified as poorly graded according to the Unified Soil Classification System, ASTM D 2487-06 [70]. 

The activator that was used was a mixture of sodium hydroxide (NaOH) (10 M) and sodium silicate (Na_2_SiO_3_) in suitable proportions (1:2, respectively). Table 4 shows the composition of the pastes made with MKW from MetaMAX and MKB from ARGECO. Each mixture was used in the construction of six specimens with dimensions of 40 × 40 × 160 mm^3^. The laboratory already had some knowledge about these mixtures [71]. All compositions were produced in accordance with the EN 196-1 standard [72]. Each mixture was poured into a 40 × 40 × 160 mm^3^ steel mold, and a vibrating table was used to remove air from the mixture. The specimen was then wrapped in plastic film and allowed to cure.

### 3.2. Curing Temperature Modeling

At the start of the process, it was decided to allow the components of each mixture to approach the curing temperature. Thus, the molds, MK, and sand were held at the desired temperature for approximately 5 to 6 h before mixing. After mixing, they were held at the study temperature for 72 h. It is during this period that almost all of the increases in the material’s strength properties occur. Afterward, they were held in an acclimatized place at the study temperature.

Relevantly, difficulties were encountered in the mixing phase when mixing at temperatures of 40 and 50 °C. The difficulties arose from the fact that the sand, MK, and molds had been previously placed at the temperatures under study. Because these elements had been held at a higher temperature, the mixture lost some of its liquid phase, thus making the mixture less fluid and more viscous. In the case of the MKW mixture, which was held at 50 °C, it was not even possible to add all of the desired binder because the mixture was very viscous and almost unworkable. Approximately 742 g of binder was added, 8 g less than intended.

In order to record the evolution of the curing temperatures of each mixture, a thermocouple was used, connected to a Data Logger. The graph of the temperatures recorded in the first days of curing at 10 °C is presented in Figure 1. The average temperature, in the first 56 h, was 10.4 °C, very close to the desired temperature of 10 °C, which was programmed in the climatic chamber. After the initial disturbance corresponding to the opening of the door, the maximum temperature fluctuation was ±1 °C.

The record of the temperature changes experienced by the specimens cured at 15 °C was lost on a pen-drive. The desired temperature was programmed, so in principle, it was achieved with only small deviations, as curing was carried out in the same climatic chamber used for the specimens cured at 10 °C.

Regarding the specimens cured at 30 °C, 40 °C, and 50 °C, for which an oven was used, Figure 2, Figure 3 and Figure 4 show the graphs of the recorded temperatures. The average temperatures to which the specimens were exposed in the first 56 h were, respectively, 30.8 °C, 40.2 °C, and 52.5 °C. The maximum temperature fluctuations were ±3.4 °C, ±2.5 °C, and ±3.5 °C, respectively. It is also possible to verify some temperature oscillations in the first 6 h (see Figure 3 and Figure 4). However, it is thought that these initial disturbances, which resulted from opening the oven door to remove the items from the mixture and later to replace specimens after mixing, did not significantly influence the final results.

It is important to highlight the differences in the final color and texture of the specimens cured at higher temperatures (30, 40 and 50 °C), when compared to those cured at lower temperatures (10, 15 and 20 °C). Figure 5 shows a specimen cured at 50 °C (on the left) and a specimen cured at 10 °C (on the right). This difference may be a consequence of the evaporation of part of the liquid phase during the curing process.

### 3.3. Flexural Test

The analysis of the influence of curing temperature on MK-based geopolymers was based on experiments testing the mechanical strength of the geopolymers, namely the flexural (tensile) strength σ_t_ and the compressive strength σ_c_.

These assessments followed, whenever possible, the EN 196-1 standard [72]. This standard, developed for cement mortars, stipulates the use of 40 × 40 × 160 mm^3^ prismatic specimens and describes a method for determining the flexural and compressive strengths. Furthermore, the EN 12390-1 standard [73], which regulates the strength tests of concrete specimens, was also followed.

For the flexural test, six specimens per mixture were tested in order to determine σ_t_. The load model presented in the EN 12390-5 [74] standard was adopted. In this model, the specimens are subjected to two actions at thirds of the span. This test was carried out in a Servosis ME-402 series hydraulic press programmed to apply a vertical deformation and carried out in two stages. The first stage took the form of a preload up to F ≈ 200 N (about one tenth of the breaking load). After that, pressure was applied at a deformation rate of 0.003 mm/s until the specimen broke.

All tests were carried out between the 15th and 17th days because the specimens are expected to have reached nearly their maximum strength capacities by that point.

### 3.4. Compressive Test

The two prisms resulting from the previous test were tested for compression in the same press, using appropriate and standardized equipment. Like the previous test, this test was carried out in two phases, with an initial small preload of up to 5000 N. Immediately afterwards, the test was run until the test piece broke, with a deformation rate of 0.01 mm/s. The EN 12390-3 [75] standard was adopted.

## 4. Results and Discussion

### 4.1. Flexural Strength

All specimens were initially submitted to the flexural test. It was possible to record the evolution of the force as the press caused deformation. It was important to record only the ultimate force. As would be expected, the rupture almost always occurred in the central zone of the specimens. Each specimen was thus approximately subdivided into two half prisms.

For MKB and MKW specimens, Appendix A displays the σ_t_ values, the average value, the standard deviation and the deviation. We will first comment on the stress variation coefficients: with the exception of the first set of specimens, all have values below 5%, indicating that the results obtained are of excellent quality.

Figure 6 shows the bar graph corresponding to the average values of σ_t_, as well as the upper and lower limits of the results. The brown areas represent the values for MKB mixtures, while the white areas represent the MKW mixtures. In an initial phase, up to curing temperatures of 20 °C, no significant variations in this strength parameter are perceptible. After this point, there is an evident tendency to breakdown, as can be seen from the strength of materials cured at 30 °C. The flexural strength is reduced by more than 45% at curing temperatures of 40 °C and 50 °C. The MKW mixtures follow the same trends as the MKB mixtures; materials cured at 50 °C, show a breakdown of about 38% compared to the materials cured at 20 °C.

The graph in Figure 7 shows the decrease in strength of the MK-based geopolymers when the curing temperature is higher than the ambient temperature in the first hours of curing. Considering these results, it is not to be expected that the flexural strength will increase at curing temperatures above 50 °C. This decline in strength aligns with the key findings outlined in Section 2.4. Likely, this reduction is associated with the higher porosity of geopolymers cured at elevated temperatures, which can be attributed to the premature setting that accompanies the accelerated geopolymerization process.

### 4.2. Compressive Strength

The curing conditions and curing time of the specimens are those mentioned in the previous section, as the compression failure test is carried out with the “halves” obtained from each specimen during the flexural test.

Figure 8 shows the double-pyramid shape of the broken specimens. This shape can be classified as satisfactory according to the EN 12390-3 [75] standard.

The compressive strength (σ_c_) values of specimens are presented in Appendix B. The excellent quality of the results is clear from the coefficients of variation. Figure 9 shows the averaged σ_c_ values, along with upper and lower limits. Conclusions similar to those drawn from the previous results (σ_t_) can be drawn here. At curing temperatures up to 20 °C, there are no significant variations, but at higher curing temperatures, there is a significant reduction in compressive strength. At curing temperatures of 40 °C and 50 °C, there is a reduction in compressive strength of over 60%. This change is also observed in MKW mixtures.

The graph in Figure 10 shows a sharp decrease in the compressive strength of MK-based geopolymers when the curing temperature in the first few hours is higher than the ambient temperature. As before, it is not expected that the compressive strength will increase for curing temperatures above 50 °C. As discussed in the context of flexural strength, the decrease in compressive strength is likely attributable to an increase in porosity.

### 4.3. Tensile Stress-Strain Diagram

For computational mechanics, knowledge of the stress-strain behavior diagrams of materials is fundamental to performing any mathematical analysis of a structure subjected to an action. By way of example, from flexural tests, and only in the context of materials cured at ambient temperature, Figure 11 presents several tensile stress-strain diagrams. The MKB2 diagram concerns MKB sample 2; the MKW9 diagram concerns MKW sample 9. The following diagrams are additionally presented: (i) The MKW diagram, as determined by Lopes [71], compared to a mixture equal to MKW; (ii) The mortar diagram, compared to an OPC mortar (see Table 5), as evaluated by Lopes [71]; (iii) the mortar v2 diagram, which contains a comparison to another mortar (see Table 5), as determined by the authors in a previous work.

The MKW and mortar diagrams shown in Figure 11 were derived using bonded strain gauges. The remaining diagrams were derived (only) from the point in gray, assuming a given value of the modulus of elasticity (MOE). In this regard, it is worth adding that the presented MOE values have a scientific basis: (i) for the MKW geopolymer, MOE = 12 GPa, as determined by Lopes [71]; (ii) for the MKB geopolymer, MOE = 18 GPa, as determined by Lopes [69]; (iii) the EC2 [39] was used to estimate the expected value for the OPC mortar given the compressive strength (MOE = 41 GPa).

Table 5 presents the composition of the mortars (C stands for cement, S for sand, and W for water), the compressive (σ_c_) and tensile (σ_t_) strengths, the values of the modulus of elasticity (MOE), the effective loading rate (dσ_t_/dt) in the second half of the diagram and the strain at ultimate stress (ε_max_). It is important to remember that all tests were carried out with deformation control. Instead of the strain rate, the loading rate is indicated because this format is customary. Regarding this point, note that EN 196-1 [72] specifies a loading-rate value of dσ_t_/dt~117 Pa/s for a three-point loading method. In these mortars, the effective loading rate was around 25% of that value. Some values related to the flexural tests of MK specimens are presented in Table 6.

Consider that the effective MOE values can vary (up to 10%) from item to item. Therefore, estimated values are average values only. For example, in the set of tests performed in Lopes [71], MOE was found to be 12.0 GPa; in the MKW [71] example shown, MOE = 12.7 GPa.

In Figure 11, the main feature that stands out is the “almost” perfect elastic behavior, which holds until brittle failure. In other words, the “stiffening effect”, characteristic of reinforced concrete elements, is not seen in this type of test. Three test characteristics are shown in the diagrams: (i) it is a bending test; (ii) the height of the specimens is reduced; (iii) the test includes a programmed (vertical) deformation rate. In fact, as this test is a flexural “tensile” test (i), on an element of very small height (ii), failure occurs on a very small surface of width b and almost infinitesimal height. From that point on, the specimen is no longer able to rebalance itself, in part due to the release of energy accumulated in the Neoprene plates used in the test. The deformation rate (iii) is the third factor contributing to the specimen’s inability to rebalance itself. At that deformation rate (3 μm/s in the test machine), the axial strain tends to grow at a rate of about 2.5 μ/s. For those reasons, this test, using this type of specimen, should not be used to learn about this very important characteristic behavior (the stiffening effect) of materials. However, the test does allow evaluation of the flexural strength and the behavior (almost) up to the point of failure.

The previous results show that the MKB geopolymer has almost double the strain of the OPC mortar at ultimate tensile stress (ε_max_). This difference is directly related to the level of flexural strength and the MOE values. Similarly, the MKW geopolymer has almost triple the values of the OPC mortar. These low MOE values have a direct impact on deflection control checks in the common serviceability limit states. It is important to consider the long-term impact of phenomena such as concrete shrinkage and creep, which can easily increase the instantaneous values of deflection in a beam or slab by three or more times. An effective creep coefficient of not less than two is sufficient to address this issue.

These stress-strain diagrams cannot be used directly in static computational analyses. The tests carried out were only semi-static, even though the loading rate was only 25% of the value specified in the EN 196-1 [72] standard. The loading rate is an important parameter to consider and is often ignored when results are presented in the literature. In this case, the loading rate was equivalent to causing the specimens to collapse in about 180 s.

### 4.4. Compressive Stress-Strain Diagram

Figure 12 and Table 7 present the results of compression tests. Figure 12 shows three diagrams: (i) “Mortar v2” represents the second mortar indicated in Table 5; (ii) “MKB2” represents one of the splits of MKB geopolymer sample 2; (iii) “MKW8” represents one of the splits of MKW geopolymer sample 8. The diagrams were determined using the following values for the MOE: 31 GPa for “Mortar v2”, 18 GPa for “MKB2”, and 12 GPa for “MKW8”. Table 7 provides additional results for the tested specimens.

For the mortar, the loading rate in the (almost) linear part of the diagram was 210 Pa/s and the strain at ultimate stress was 1377 μ. This loading rate represents only ~14% of the average value recommended in standard 196-1 [72] (1500 Pa/s) or ~35% of the average value recommended in standard 12390-3 [75] (600 Pa/s). Even so, this quasi-static test yielded quasi-linear diagrams up to the point of ultimate stress. This result is similar to that reported by Ouyang [76]. However, it is known that for concrete, some loss of stiffness is expected starting from 40% of the ultimate stress [40]. Furthermore, for mortar, the strain at peak stress is also less than 2.5‰. Relatedly, Bischoff and Perry [77] report that for this strain rate level (~7 μ/s), dynamic effects are negligible in the evaluation of ultimate stress but not at the level of strain values at ultimate stress.

As shown in Figure 12, and Table 7, the strain values at ultimate stresses of the MKB geopolymer are substantially higher (~45% more) than those of the mortar. The MKW geopolymer has more than double (2.25×) the value of ε_max_ compared to the mortar.

## 5. Conclusions

This study aims to investigate whether an increase in temperature during curing of large masses of MK geopolymers can impact their strength characteristics, as assessed in tests using small specimens. Therefore, achieving the main objective of this experimental work involved the evaluation of the mechanical properties of geopolymeric mixtures by varying the curing temperature to which the specimens were subjected. Samples were constructed and subjected to curing temperatures of 10, 15, 20, 30, 40, and 50 °C for about 50 h.

For curing temperatures above 30 °C, flexural strength decreased significantly. For the metakaolin ARGECO-based geopolymer, the losses were on the order of 45% for a curing temperature of 50 °C compared to the ambient temperature. This value was 38% for the metakaolin MetaMax-based geopolymer. This value represents a very significant loss, which is concerning for mixtures cured under these conditions. The loss of compressive strength compounded this pattern, with losses of up to 62% (!).Mixtures made up of white (MetaMax) and brown (ARGECO) metakaolin were also compared, and important conclusions can be drawn from the results. Regarding the flexural strength, the mixtures of brown metakaolin cured at room temperature were significantly superior. In the analysis of the compressive strength, the opposite result was found, with the mixtures constructed with the white metakaolin having superior strength. However, these differences are always below about 7% in average terms.Based on the behavior diagrams representing the metakaolin-based geopolymers, it can be concluded that their deformation at ultimate stress is substantially higher than that of an equivalent OPC mortar, both in the flexural test and in the compression test. Compared to the mortar, in the flexural tests, the strain at ultimate stress of the white and brown geopolymers are about 2.7× and 1.9× greater, respectively. In the compression tests, the corresponding values are 2.3× and 1.5× greater, respectively. It is important, however, to remember that these strain values correspond to single short-term loading. Over the long term, only the effects on concrete (shrinkage and creep) are known.

The objective of this study is to enhance the practicality of utilizing geopolymers made from alkaline activation of MK as an alternative to concrete for structural applications. Therefore, this research serves as an exploratory investigation of the mechanical characteristics of geopolymers. Importantly, it has been observed that these materials do not require additional energy for curing after mixing, aligning their production with the goal of producing environmentally friendly materials.

Considering the main question posed in the introduction, predicting a temporal temperature diagram for a specific MK geopolymer appears challenging. Factors such as formwork, mass quantity, and other considerations would need to be taken into account. However, several questions arise from this study, and some of them are particularly noteworthy. One interesting area for future research is the behavior of geopolymer materials in the presence of water (polluted or not). This problem is especially important for the use of these materials in countries with limited resources. Additionally, a study on the economic feasibility and environmental impact of manufacturing structural elements using geopolymer materials would be valuable. Developing methodologies for the preservation of the liquid phase under adverse curing conditions, such as high temperatures, is another important consideration. Finally, it would be beneficial to create additives similar to those used in concrete that could enhance the workability of the pastes and potentially increase their strength.

## Figures and Tables

**Figure 1 materials-16-07460-f001:**
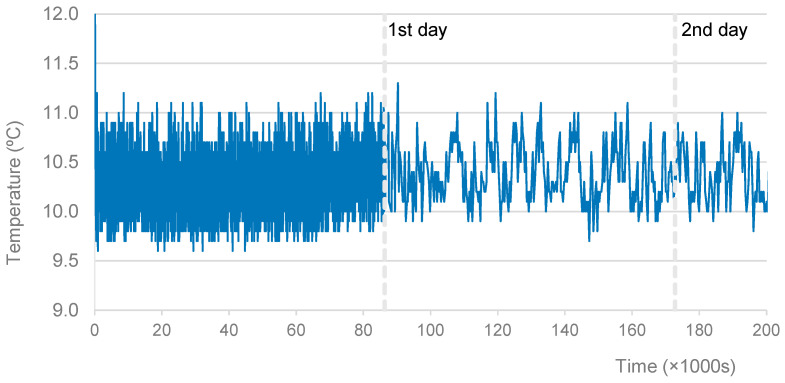
Specimens cured at 10 °C. Record of curing temperatures.

**Figure 2 materials-16-07460-f002:**
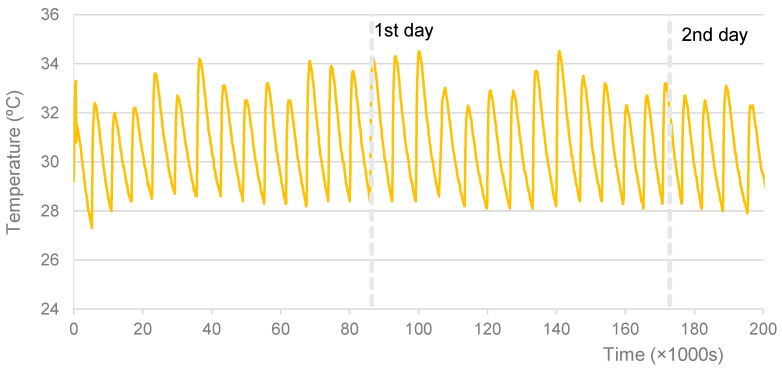
Specimens cured at 30 °C. Record of curing temperature.

**Figure 3 materials-16-07460-f003:**
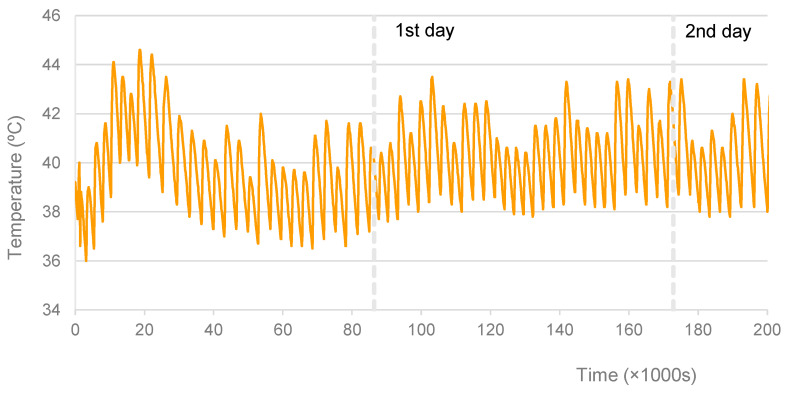
Specimens cured at 40 °C. Record of curing temperature.

**Figure 4 materials-16-07460-f004:**
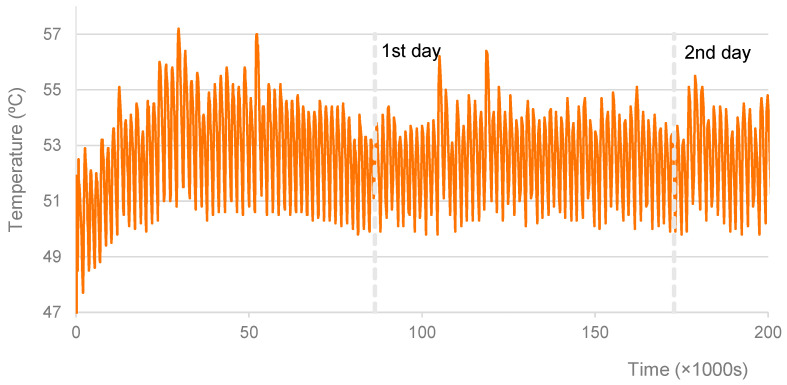
Specimens cured at 50 °C. Record of curing temperature.

**Figure 5 materials-16-07460-f005:**
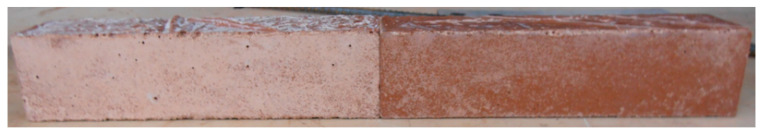
Feature of specimens manufactured at different curing temperatures (50 °C and 10 °C).

**Figure 6 materials-16-07460-f006:**
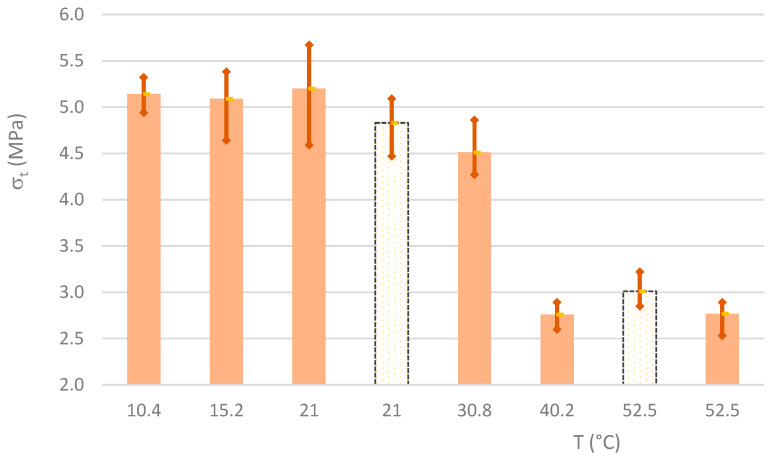
Average and extreme values of flexural strength.

**Figure 7 materials-16-07460-f007:**
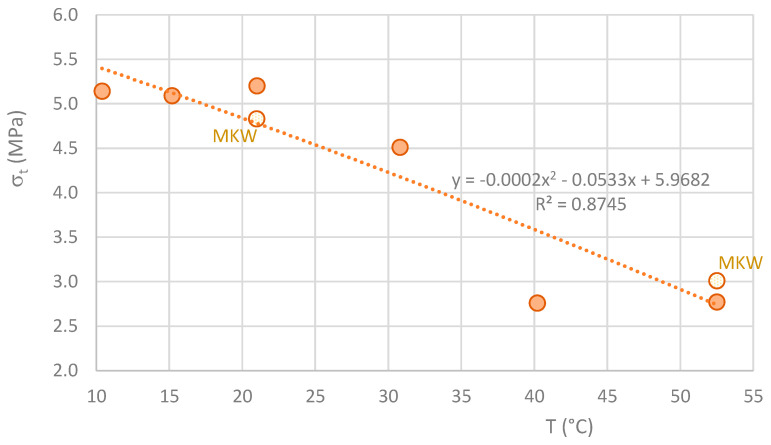
Variation of the averaged values of flexural strength with temperature.

**Figure 8 materials-16-07460-f008:**
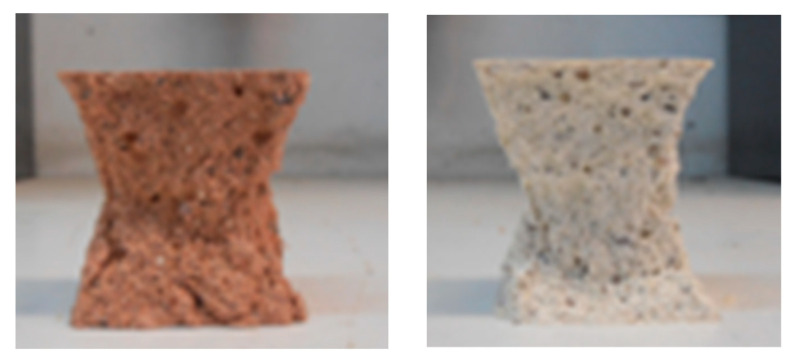
Type failures of specimens submitted to the compression failure test.

**Figure 9 materials-16-07460-f009:**
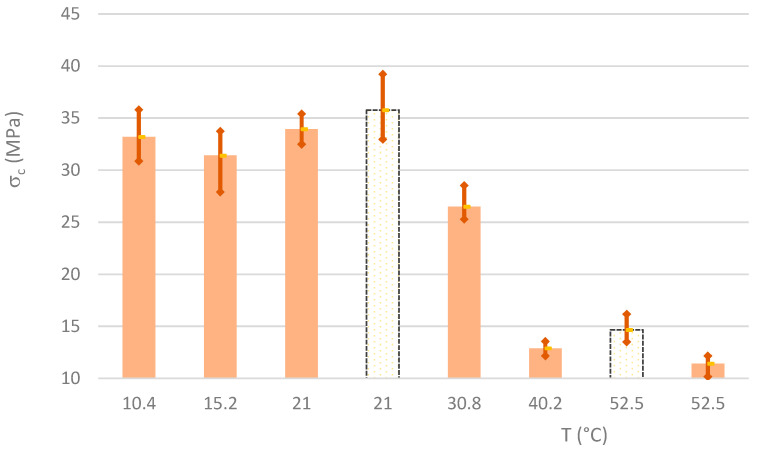
Average and extreme values of compressive strength.

**Figure 10 materials-16-07460-f010:**
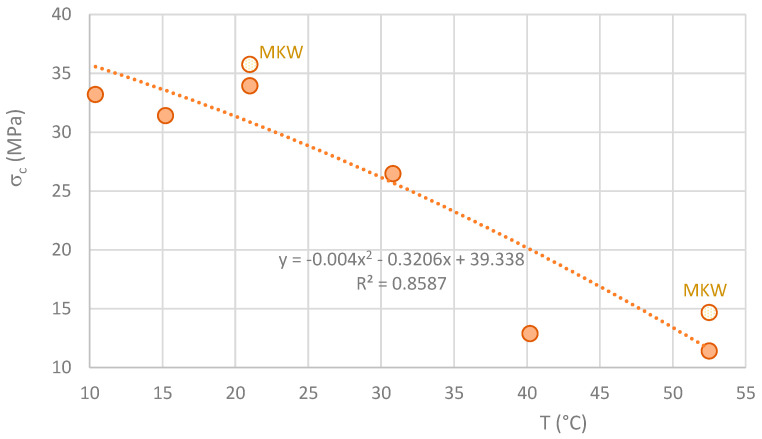
Variation of the averaged values of compressive strength with temperature.

**Figure 11 materials-16-07460-f011:**
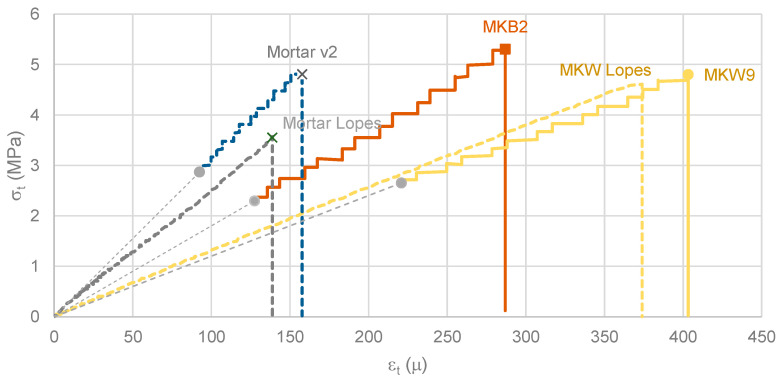
Tensile stress-strain diagrams; Ambient curing temperature.

**Figure 12 materials-16-07460-f012:**
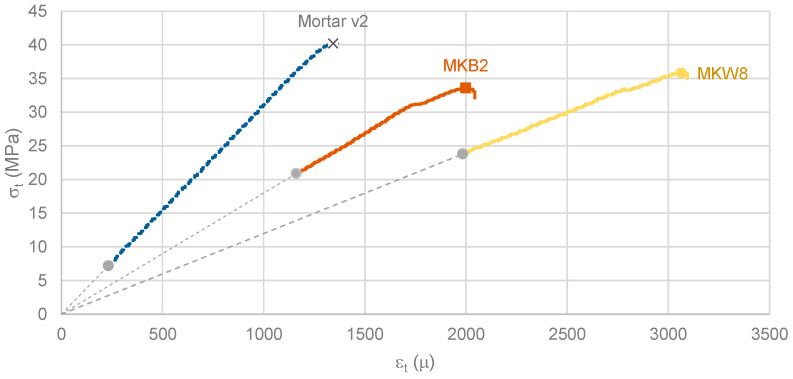
Compressive stress-strain diagrams; ambient curing temperature.

**Table 1 materials-16-07460-t001:** References on shrinkage values.

Reference	Material	Age (Days)	Shrinkage
Duran et al. [42]	OPC mortar (w/c = 0.5)	90	DS~1‰
Duran et al. [42]	GGBFS	90	DS~3–6‰
Xiang et al. [7]	GGBFS	90	DS~3–6‰
Yuan et al. [6]	GGBFS	10 28	DS~0.4‰; DS~0.45‰
Maochieh Chi [43]	GGBFS	28	DS~0.4–0.8‰
Ling et al. [44]	Class C FA	56	DS~6.2‰
Ma and Ye [45]	Class F FA	28	DS~0.5–2.5‰
Saha and Rajasekaran [46]	FA	28	TSD~1.2–2.5‰
Zhang [47]	MK	--	DS~4–6‰
Yang et al. [48]	MK	--	DS~4–6‰
Campopiano et al. [49]	MK	--	DS~2–4‰
Palumbo et al. [50]	MK	--	DS~2–4‰
Chen et al. [51]	MK	--	DS~2–4‰
Xiang et al. [7]	MK	--	DS~2–4‰
Mobili et al. [52]	MK	--	DS~1–2‰
Tonnayopas et al. [53]	MK	~100–200	DS~0.5–0.6‰
Trincal et al. [54]	MK	--	DS~1–2‰
Eisa et al. [19]	MK	--	DS < ~0.6‰

**Table 2 materials-16-07460-t002:** References on application of curing temperature to MK geopolymers.

Reference	Curing Temperature	Composition	Compressive Strength	Main Conclusion
Lahale [63]	10 °C, 20 °C, 30 °C (during 3 days)	Mortar	~58 MPa (at 7 day)	Reduction of 4% at 30 °C
Mo [64]	20 °C, 40 °C, 60 °C, 80 °C, 100 °C (until tested)	MK + Alkaline solution	~100 MPa (at 7 day)	Reduction at day 3 if temperature is greater than 80 °C; reduction at day 7 if temperature is greater than 60 °C;
Li [65]	Ambient, 60 °C, 90 °C	MK + Alkaline solution and mortar	~60–70 MPa (at 7 day)	Reduction at day 7 if temperature is greater than 90 °C
Rovnanik [66]	10 °C, 20 °C, 40 °C, 60 °C, 80 °C (during 1–4 h)	Mortar	~50 MPa (at 28 day)	Greater reduction if time at a given temperature increases
Arellano [67]	Ambient, 75 °C (during 24 h)	Mortar	~50 MPa (at 14–28 day)	Reduction of ~10% if temperature is 75 °C
Vitola [68]	80 °C (during 24 h)	MK + Alkaline solution	≈5–25 MPa	--

**Table 3 materials-16-07460-t003:** Elemental analysis of MetaMax MKW and ARGECO MKB metakaolins.

Elements		SiO_2_	Al_2_O_3_	TiO_2_	MgO
MetaMax	MKW	57%	37.0%	2.73%	1.26%
ARGECO	MKB	70.2%	21.8%	2.2%	1.15%

**Table 4 materials-16-07460-t004:** Mixture composition of the MK-based geopolymer.

Mixture	MK	Sand	Activator	NaOH	Na_2_SiO_3_
MKW	750 g	1875 g	900 g	300 g	600 g
MKB	750 g	1875 g	645 g	215 g	430 g

**Table 5 materials-16-07460-t005:** Composition, strength and flexural-test characteristics of mortar specimens.

Mortar	C:S:W	σ_c_ (MPa)	σ_t_ (MPa)	MOE (GPa)	dσ_t_/dt (Pa/s)	ε_max_ (μ)
Mortar [71]	1:4.43:0.67	21.7	3.56	28.3	26.6	~138
Mortar v2	1:4.5:0.5	40.7	4.06 *	31 *	32.6	158

(*) Estimated value.

**Table 6 materials-16-07460-t006:** Flexural-test characteristics of MK specimens.

Item	dσ_t_/dt (Pa/s)	ε_max_ (μ)	Item	dσ_t_/dt (Pa/s)	ε_max_ (μ)
1	45.0	301	7	32.1	386
2	42.9	287	8	31.2	427
5	38.0	264	9	34.4	413
6	35.9	274	10	31.2	375

**Table 7 materials-16-07460-t007:** Compressive test characteristics of MK samples.

Item	dσ_c_/dt (Pa/s)	ε_max_ (μ)	Item	dσ_c_/dt (Pa/s)	ε_max_ (μ)
1	122	2111	7	205	3218
2	115	1998	8	177	3064
5	115	1850	9	190	3017
6	123	1946	10	156	3097

## Data Availability

Data are contained within the article.

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
