# Peer review of "Influence of Curing Temperature on the Strength of a Metakaolin-Based Geopolymer"

_materials, 2023, doi:10.3390/ma16237460_

Round 1
Reviewer 1 Report (New Reviewer)
Comments and Suggestions for Authors
The manuscript, entitled "Influence of curing temperature on the strength of a metakaolin-based geopolymer," presents an interesting experimental study related to the influence of curing temperature on the properties of geopolymers. However, the introduction section is general, and the novelty of the paper wasn't presented. The paper needs major revisions before it is processed further; some comments follow:
Abstract: This section must be suitable for separate presentations (independent of the manuscript text body). Please present the novelty of the paper.
Introduction and Initial Considerations Section
The introduction section can be improved. Please introduce a corresponding citation to back up your affirmations from lines 30-47, etc.
Also, please present relevant literature in quantitative form. Include clear values for the tested properties and the effect of key parameters on those properties.
What is the novelty of this study? Please clearly highlight what makes the results from this research different from those previously reported and how this paper will extend the knowledge in this field.
The discussion part of this manuscript mainly analyzes and compares the experimental data before and after, but does not discuss the results by adding available reasoning. I suggest that the author add some reasoning on the basis of the literature and the possible reasons for this situation.
Although the authors evaluated or discussed relevant studies, the introduction or literature review could be improved by introducing a few tables or schematic representations that can summarize the parameters and results from the reviewed literature.
Please merge these two sections into one short and relevant presentation of the literature in the field of the aim of this research.
Materials and methods
Why do the authors consider the parameters presented in Table 2 to be the most important? What was the rationale for choosing these parameters? Please briefly describe the cause-and-effect relationship between each chosen parameter and improve the description of the section accordingly. Please cite corresponding studies that confirm those relationships.
How many samples have been tested from each batch? The repeatability and credibility of the results are strongly related to the number of tests performed.
Results and discussions
Figure 8: Please introduce a scale bar on the figure and provide images with higher resolution. The images are blurred.
Discussion section. The discussion section is missing. In the discussion section, a clear correspondence and comparison between the results of this study and those in the literature should be provided. Please improve. Currently, the discussion section includes some comments and appreciation about the obtained results without any comparison with the literature.
Please highlight the differences between the previously obtained results and the results obtained by the authors considering traffic and sustainable materials.
Conclusions section: Please improve the conclusions and present them following the main recommendations by academia of giving the conclusions of the study in points with highlights.
Future directions and limitations: Please provide some future directions and limitations of the study. This section is very important for this study because some strict or limited parameters were considered (temperatures, types of raw materials, mixtures, etc.).
Please check the paper for typing errors. CO2 in line 35, etc.
mm3 in line 383, etc.
Author Response
Please see the attachment

Reviewer 2 Report (Previous Reviewer 4)
Comments and Suggestions for Authors
materials-2566761
Influence of curing temperature on the strength of a metakaolin-based geopolymer
The study on metakaolin-based geopolymers provides valuable insights into their mechanical properties under varying curing temperatures. However, a major revision is needed to improve clarity and coherence. The content is challenging to follow, and the findings need to be presented more effectively. Technical terms and abbreviations need explanation, and the conclusion section lacks a clear synthesis of key findings. Addressing these concerns will enhance the study's impact.
Comments:
- How do the flexural and compressive strengths of metakaolin-based geopolymers vary with different curing temperatures (10, 15, 20, 30, 40, and 50 °C), and what implications does this have for their use in civil construction?
- The observed reduction in flexural strength at curing temperatures above 30 °C is significant. What mechanisms are responsible for this decline in strength, and how does it differ between brown (AR-GECO) and white (MetaMax) metakaolin-based geopolymers?
- In the comparison between brown and white metakaolin-based geopolymers, the flexural strength is better for brown metakaolin at room temperature, while compressive strength favors white metakaolin. What factors contribute to this divergence in performance between the two types of geopolymers?
- The study emphasizes the importance of curing temperatures below 30 °C to avoid strength reduction. What practical challenges might be faced in real-world construction scenarios to maintain such controlled curing conditions, and how could these challenges be addressed?
- The stress-strain diagrams provided for metakaolin-based geopolymers in flexural tests indicate almost perfect elastic behavior until brittle failure. How might this behavior influence the structural performance and durability of geopolymers in real-world applications?
- The loading rate in the compression tests is mentioned as an important parameter, and it's noted that it's often ignored in the literature. How does the loading rate influence the results, and why is it crucial to consider in the context of these tests?
- The study mentions the need for long-term considerations, specifically regarding concrete shrinkage and creep. How might geopolymers, as an alternative to concrete, address or exhibit resistance to these long-term challenges?
- The research highlights the potential environmental benefits of geopolymers, given that they do not require additional energy for curing. Could authors elaborate on the environmental impact and sustainability of geopolymer production compared to traditional concrete?
- Considering the observed losses in flexural and compressive strengths at higher curing temperatures, are there specific applications or scenarios where metakaolin-based geopolymers might still be advantageous despite these limitations?
- The study recommends further research on the behavior of geopolymers in the presence of water. What specific aspects of water interaction (polluted or not) are crucial to understanding the performance of geopolymers, and how might these findings impact real-world applications?
- In the context of economic feasibility, what are the potential cost implications of using metakaolin-based geopolymers in comparison to traditional concrete, and are there any notable economic advantages or challenges?
- The study suggests the need for additives similar to those used in concrete to enhance workability and potentially increase strength. What types of additives could be considered, and how might they affect the overall performance of metakaolin-based geopolymers?
- Preservation of the liquid phase under adverse curing conditions, such as high temperatures, is mentioned as an important consideration. What strategies or methodologies could be explored to achieve this preservation, and how might it impact the practical application of geopolymers?
- Could the observed strain values at ultimate stress be indicative of potential issues or advantages in real-world applications, especially in comparison to traditional concrete, considering both short-term and long-term loading scenarios?
- Given the focus on environmentally friendly materials, are there any potential by-products or waste generated during the production of metakaolin-based geopolymers, and how might these be managed to ensure overall environmental sustainability?
- Address improper reference errors by eliminating instances of "Error! Reference source not found." in the text.
Minor editing of English language required.
Round 2
Reviewer 1 Report (New Reviewer)
Comments and Suggestions for Authors
Dear author,
You have done a great job in revising the paper.
Best regards,
Reviewer 2 Report (Previous Reviewer 4)
Comments and Suggestions for Authors
Manuscript ID: materials-2566761R1
Title: Influence of curing temperature on the strength of a metakaolin-based geopolymer
The authors satisfactorily addressed all of my comments. The revised manuscript, titled "Influence of Curing Temperature on the Strength of a Metakaolin-Based Geopolymer," demonstrates commendable research work and is suitable for publication.
This manuscript is a resubmission of an earlier submission. The following is a list of the peer review reports and author responses from that submission.
Round 1
Reviewer 1 Report
Comments and Suggestions for Authors
The article entitled “Influence of curing temperature on the strength of a metakaolin-based geopolymer” has been evaluated.
The Abstract and Introduction have been rewritten and new theoretical passages have been added to introduce the reader to the issues. The results were only extended with tensile stress and compressive stress-strain diagrams.
However, none of my previous comments were accepted (IR, XRD analyses, strengths determined over a longer time frame, e.g. 28 days) and an article based solely on mechanical properties is not sufficient. Although the authors cure the samples at different temperatures, they have not confirmed by any analysis whether new crystalline phases are formed.
Moreover, the text is full of errors (indexes, missing brackets, poorly defined/missing references). I honestly don't understand how anyone can submit a manuscript in such a state.
I still think that the quality of the paper is low, there is a lack of originality and the data presented are not sufficient for publication in Materials.
Therefore, I do not recommend the publication.
Reviewer 2 Report
Comments and Suggestions for Authors
The research article titled: Influence of curing temperature on the strength of a metakaolin-based geopolymer, by Lopes have revised the article very well. Here are just few suggestions before being publish;
1. Write the abstract in present tense.
2. There is no need to mention the websites in lines 41-42.
3. Please be careful about the subscript and superscripts.
4. I suggest the author to highlight some recent article like Journal of Materials in Civil Engineering 28, no. 7 (2016): 04016035.
Comments on the Quality of English LanguageEnglish is good.
Reviewer 3 Report
Comments and Suggestions for Authors
The manuscript has been well revised, and all issues questioned in the first review round has been well polished. The revised manuscript is recommended to be accepted for publication.
Reviewer 4 Report
Comments and Suggestions for Authors
materials-2503192
Comments: The manuscript needs major revision, below are suggestion that needs to be addressed by the authors in the revised manuscript.
Manuscript Revision Suggestions:
Added New Section (after Introduction): Geopolymer Mechanism
1. The manuscript requires major revisions. It is crucial to extensively address the polymerization mechanism in geopolymer as it governs the material's physical and mechanical properties. Understanding the chemical reactions and polymerization process involved is essential. For a more comprehensive and in-depth discussion of this topic, the following articles can be referenced: doi.org/10.1016/j.cscm.2023.e02133; doi.org/10.1016/j.clay.2023.107020; doi.org/10.1016/j.matlet.2023.134784.
From Section 2.3: Shrinkage
1. What are the three components of strain experienced by hardened concrete over time under compressive stress or load-free conditions?
2. How do geopolymers made from waste or industrial by-products differ from normal concrete in terms of total shrinkage deformation (TSD)?
3. What is the typical range of total shrinkage deformation (TSD) for normal concrete used in construction?
4. Why are lower values of total shrinkage deformation (TSD) not problematic in concrete construction?
From Section 2.4: Effect of Curing Temperature
5. What is the significance of defining the temperature parameter in the context of curing concrete? What temperature range is typically considered for carrying out tests in the structural field? What is the specified temperature range for curing concrete specimens according to standard EN12390-2?
6. What aspect regarding the temperature during the curing process of concrete specimens does standard EN12390-2 ignore? How does standard EN12390-2 define the specified temperature value for curing concrete specimens?
From Section 3.2: Curing Temperature Modeling
7. Why was it decided to allow the components of each mixture to approach the curing temperature before mixing?
8. How long were the molds, metakaolin, and sand left at the desired temperature values before? During which period was the maximum increase in the material's strength recorded?
9. What difficulties were encountered during the mixing phase when the temperatures were 40 and 50 °C?
10. How was the evolution of the curing temperatures of each mixture recorded?
From Section 4.2: Compressive Strength
11. How are the specimens used for the compression failure test obtained?
12. How is the rupture format of the specimens classified according to the EN 12390-3 standard?
13. What conclusions can be drawn about the compressive strength (σc) of the specimens based on the results?
14. How does the compressive strength of the specimens change with higher curing temperatures?
15. What does Figure 10 depict regarding the compressive strength of MK-based geopolymers?
From Section 4.4: Compressive Stress-Strain Diagram
16. What do the diagrams in Figure 12 represent?
17. What values were used for the modulus of elasticity (MOE) in determining the diagrams in Figure 12?
18. What was the loading rate used during the compression test for the mortar case?
19. How does the quasi-static test for mortar differ from the behavior typically observed in concrete?
20. According to Bischoff and Perry, what is the significance of the strain rate level in evaluating the ultimate stress in mortar?
Comments on the Quality of English Language
Minor editing of English language required.
Reviewer 5 Report
Comments and Suggestions for Authors
SUMMARY
The article submitted for review is relevant. It studied the influence of curing temperature on the strength of a metakaolin-based geopolymer. The relevance of the study is due to the need to study the use of artificially agglomerated stone materials in old buildings, which have demonstrated high strength and durability. It has been established that these agglomerated materials were the first artificial compounds based on alkaline activation. In their study, the authors develop a new family of geopolymer materials obtained by alkaline activation of the binder. The authors want to solve the problem of an alternative to concrete for use in civil engineering. The authors are working to propose new practical developments of materials for real structures, namely for beams and slabs of buildings. They obtained a number of important results, which makes it possible to consider their study as practically significant and scientifically new. At the same time, the reviewer had a number of comments, they are presented below.
COMMENTS
1) Authors are encouraged to more clearly formulate the scientific problem in the abstract. The authors state that they have been developing an alternative to concrete for the construction of civil buildings, but it needs to be shown what the problem is with the use of concrete in civil buildings. Perhaps this problem should be linked to the need to comply with the environmental sustainable development agenda or some other reasons, so that it is obvious to the reader why the study was carried out.
2) The abstract looks overly detailed. Perhaps it should be somewhat concretized and brought to the form: a scientific problem, a description of the methodology, a scientific result, a quantitative and qualitative description of the results.
3) The section "Introduction" is not presented in sufficient detail to talk about the analysis of the current state of the issue of geopolymers based on metakaolin. There are a lot of studies on this topic, and in general, the science of geopolymers is actively developing in different countries. But the authors presented a review of only 19 references, which does not fully reflect the current state of this problem. Authors should deepen their literature review, and their section should end with a clear statement of the scientific problem, as well as the purpose and tasks of the study.
4) The authors give section 2 in a rather unusual form, calling it "Initial Considerations". It is likely that such fragments will be useful in section 1 "Introduction" in the same place where the literature review is given. The authors provide a lot of information and reach a review of 68 references. This causes a fairly positive impression on the reviewer. But it seems that the authors should structure their research. Perhaps the descriptions of some links should reduce the amount of information provided. Still, the authors do not declare their study as a Review. They do their own research. Here, probably, such a volume of information about previous studies will be redundant. Authors are encouraged to work on Section 2.
5) The section "Materials and Methods", presented in section 3, at the same time, is not detailed enough. Authors should justify the choice of materials for the study and describe the methodology in more detail.
6) Figure 1 seems interesting, but it is poorly interpreted in the text. The same remark applies to figures 2-4.
7) I would like to see the photo in Figure 5 in a higher quality. The graph in Figure 7 seems not very informative, detailed explanations should be given on the figure.
8) The photographs in Figure 8 are of inadequate quality. You should replace the picture with a high-quality one.
9) The dependencies in Figure 12 are unclear, they should be explained.
10) Tables 1 and the table after it need a detailed explanation, they look unnecessarily cumbersome.
11) In general, there are a lot of technical typos and inconsistencies in the text. Authors should carefully proofread and edit their article.
12) Discussion of the obtained results is not presented enough. A detailed comparison of the obtained results with the results previously obtained by other authors should be given.
13) The conclusions should be specified in terms of the new knowledge gained and the developed ideas about geopolymers based on metakaolin, what new contribution to science the authors made. This should be detailed.
14) The list of references seems to be insufficiently elaborated. There are technical flaws. Authors should finalize the list of references, arrange references properly.

Comments on the Quality of English LanguageEnglish language test required
Reviewer 6 Report
Comments and Suggestions for Authors
After reviewing the manuscript, I have no other comments, my suggestion: Accept.